

# Characterization of the Vaporization Inlet for Aerosols (VIA) for Online Measurements of Particulate Highly Oxygenated Organic Molecules (HOMs)

Jian Zhao[1,*], Valter Mickwitz[1,*], Yuanyuan Luo[1], Ella Häkkinen[1], Frans Graeffe[1], Jiangyi Zhang[1], Hilkka Timonen[2], Manjula Canagaratna[3], Jordan E. Krechmer[3,a], Qi Zhang[4,5], Markku Kulmala[1], Juha Kangasluoma[1], Douglas Worsnop[1,3], and Mikael Ehn[1]

[1]Institute for Atmospheric and Earth System Research/Physics, Faculty of Science, University of Helsinki, Helsinki, 00014, Finland
[2]Atmospheric Composition Research, Finnish Meteorological Institute, 00560, Helsinki, Finland
[3]Aerodyne Research Inc., Billerica, Massachusetts, 01821, United States
[4]Department of Environmental Toxicology, University of California, Davis, California, 95616, United States
[5]Agricultural and Environmental Chemistry Graduate Program, University of California, Davis, California, 95616, United States
[a]Now at: Bruker Daltonics Inc., Billerica, Massachusetts, 01821, United States
[*]These authors contributed equally to this work

*Correspondence to*: Jian Zhao (jian.zhao@helsinki.fi) and Mikael Ehn (mikael.ehn@helsinki.fi)

**Abstract.** Particulate matter has major climate and health impacts, and it is therefore of utmost importance to be able to measure the composition of these particles to gain insights into their sources and characteristics. Many methods, both offline and online, have been employed over the years to achieve this goal. One of the most recent developments is the Vaporization Inlet for Aerosols (VIA) coupled to a nitrate Chemical Ionization Mass Spectrometer (NO$_3$-CIMS), but a thorough understanding of the VIA–NO$_3$-CIMS system remains incomplete. In this work, we ran a series of tests to assess the impacts from different systems and sampling parameters on the detection efficiency of highly oxygenated organic molecules (HOMs) in the VIA–NO$_3$-CIMS. Firstly, we found that the current VIA system (which includes an activated carbon denuder and a vaporization tube) efficiently transmits particles (> 90% for particles larger than 50 nm), while removing gaseous compounds (> 97% for tested volatile organic compounds (VOCs)). One of the main differences between the VIA and traditional thermal desorption (TD) techniques is the very short residence time in the heating region, on the order of 0.1 s. We found that this short residence time and the corresponding short contact with heated surfaces, is likely one of the main reasons why relatively reactive or weakly bound, such as peroxides, were observable using the VIA. However, the VIA also requires much higher temperatures to fully evaporate the aerosol components. For example, the evaporation temperature of ammonium sulfate particles using the VIA was found to be about 100-150 °C higher than in typical TD systems. We also observed that the evaporation of particles with larger sizes occurred at slightly higher temperatures compared to smaller particles. Another major aspect that we investigated was the gas-phase wall losses of evaporated molecules. With a more optimized interface between the VIA and the NO$_3$-CIMS, we were able to greatly decrease wall losses and thus improve on the sensitivity compared to our earlier VIA work. This interface included a dedicated sheath flow unit to cool the heated sample and provide the NO$_3$-CIMS with the needed high flow (10 L min$^{-1}$). Our results indicate that most organic molecules observable by the NO$_3$-CIMS can evaporate and be transported efficiently in the VIA system, but upon contact with the hot walls of the VIA, the molecules are instantaneously lost. This loss potentially leads to fragmentation products that are not observable by the NO$_3$-CIMS. Thermograms, obtained by scanning the VIA temperature, were found to be very valuable for both quantification purposes and estimating the volatility of the evaporating compounds. We developed a simple one-dimensional model to account for the evaporation of particles and the temperature-dependent wall losses of the evaporated molecules, and thereby estimate the concentration of HOMs in SOA particles. Overall, our results provide much-needed insights into the key processes underlying the VIA–NO$_3$-CIMS method.



Although there are still some limitations that could be addressed through hardware improvements, the VIA-NO$_3$-CIMS is a very promising and useful system for fast online measurements of HOMs in the particle phase.

### 45    1 Introduction

Organic aerosol (OA) contributes a large portion (20-70%) to atmospheric fine particles in the lower troposphere (Zhang et al., 2007), but remains less characterized compared to other particulate components because of its vast number ($> 10^4$ organic species) of constituents (Goldstein and Galbally, 2007). OA can be directly emitted into the atmosphere through various sources (e.g. gasoline/diesel vehicles, biomass burning, and daily cooking) as primary (P)OA, and can be formed

through condensation of the oxidation products of volatile organic compounds (VOCs) as secondary (S)OA. Recently, highly oxygenated organic molecules (HOMs) were found to contribute considerably to SOA formation (Kulmala et al., 2014; Ehn et al., 2014; Jokinen et al., 2015). These OA particles greatly affect the global climate and human health (Kuniyal and Guleria, 2019; Déméautis et al., 2022) and thus need to be well characterized. The traditional techniques of OA measurement, based on filter collections followed by lab analysis, tend to be limited by low time resolution (hours to days),

sample degradation/evaporation, and incomplete analyte extraction. Thus, the development and application of online techniques, with the ability to track the formation and evolution of organic species in real-time (against rapid change in various sampling conditions), could improve our ability to understand and model OA over its entire atmospheric lifetime (Hallquist et al., 2009; Heald and Kroll, 2020).

Online mass spectrometers (MS) have been widely used to study the size-resolved chemical composition of aerosol

particles and have greatly improved our understanding of OA over the past decades, such as the single particle laser MS (Murphy, 2007) and the Aerodyne Aerosol MS (AMS) based on thermal evaporation (600 ºC) and electron impact (EI) ionization for non-refractory submicron particles (Canagaratna et al., 2007). Furthermore, the linear superposition and reproducibility of mass spectra of individual components from EI are crucial for the source apportionment of different OA factors (Zhang et al., 2011). However, the significant fragmentation introduced by these hard ionization techniques makes

the identification of parent molecules difficult, even in relatively well-controlled lab experiments. Soft-ionization, especially chemical ionization (CI), is one of the solutions to achieve near "molecular-level" measurements of gaseous organic species in trace concentrations (Huey, 2007; Zahardis et al., 2011). One advantage of soft ionization (i.e. less fragmentation) lies in the potential to identify molecular markers to track emission sources and oxidation pathways of different organic species in the gas phase. The detection of OA usually involves the vaporization of particles with

subsequent ionization of the resulting gas-phase compounds and analysis by a mass spectrometer. Temperature-programmed thermal desorption analysis could be used to obtain the volatility information of different OA components (Stark et al., 2017; Thornton et al., 2020). In addition, gas-to-particle partitioning could be investigated by deploying simultaneous gas- and particle-phase measurements, but the results need to be carefully interpreted (Stark et al., 2017; Gkatzelis et al., 2018). For example, thermal decomposition may bias the real distribution of particle-phase products and

there is the possibility that the same molecular formulas identified in both the gas and particle phases are in fact isomers (Isaacman-Vanwertz et al., 2017).

Currently widely used online "molecular-level" techniques are summarized in Table S1 and briefly described below. The Thermal Desorption Chemical Ionization Mass Spectrometer (TDCIMS), using a charged metal filament to effectively collect sub-20 nm particles, was designed to obtain the chemical composition of freshly nucleated ultrafine particles (Voisin

et al., 2003; Smith et al., 2004; Li et al., 2021). Another Aerosol CIMS technique, which thermally vaporizing particles in a heated tube without size-selecting, was deployed to detect organic molecules with different functionalities using various positive and negative reagent ions (e.g. NO$^+$, H$^+$(H$_2$O)$^2$, O$_2^-$, and F$^-$) (Hearn and Smith, 2004; Hearn and Smith, 2006), but



with relatively higher detection limit compared to other techniques. Filter Inlet for Gases and AEROsols (FIGAERO)-CIMS, using a Teflon (PTFE) filter, is capable of achieving a unit collection efficiency of particles and simultaneous gas- and particle-phase measurements down to ppt levels (Lopez-Hilfiker et al., 2014). However, potential perturbation owing to the absorption of semi- to low-volatile vapors onto the Teflon surface might be an issue (Matsunaga and Ziemann, 2010; Krechmer et al., 2016), although it can be minimized to large extents by using a pre-stage filter for blank measurement (Lopez-Hilfiker et al., 2014; Thornton et al., 2020). Without pre-collection, a Chemical Analysis of Aerosol Online (CHARON) inlet (Eichler et al., 2015) coupled with a proton-transfer-reaction (PTR) MS (Yuan et al., 2017) was used to detect VOCs and oxygenated (O)VOCs in the particle phase. The separation of particles from the gas phase is achieved by using a charcoal denuder, and this system can resolve 20-30% of the SOA mass (based on comparison to the AMS measurements) almost without thermal decomposition (Gkatzelis et al., 2018). However, the protonation-induced ionic fragmentation of oxygenated organic molecules, in particular peroxides, might be enhanced by high-energy collisions in the strong electric field of the ion drift tube. This was confirmed by both experimental studies and theoretical computations (Müller et al., 2017; Li et al., 2022; Peng et al., 2023), limiting its detection of the most oxidized species. Recently, an extractive electrospray ionization (EESI) inlet was designed to extract OA samples into charged droplets, thereby preventing thermal decomposition and ion-induce fragmentation (Lopez-Hilfiker et al., 2019). This technique was subsequently upgraded to a dual-phase version, which is now capable of measuring both gas- and particle-phase species but with different response factors (Lee et al., 2022).

The most recent addition to the suite of instrumentation used to measure aerosol components is the Vaporization Inlet for Aerosols (VIA) coupled with a nitrate (NO$_3$)-CIMS (Häkkinen et al., 2023). The NO$_3$-CIMS is routinely used to detect HOMs (i.e. the most oxidized organic species from the gas phase), which are known to be important contributors to OA in the atmosphere. Coupled with the new VIA system, the NO$_3$-CIMS has been shown to also be able to detect HOMs from particles. The paper by Häkkinen et al. (2023), as a proof-of-concept work, presents that without sample collection, continuous thermal desorption and online detection of particle-phase HOMs are feasible after removing the gaseous compounds. By coupling the VIA with a NO$_3$-CIMS, the detection limits were reported below 1 ng m$^{-3}$ for a single HOM compound (Häkkinen et al., 2023), and the relationship between the gas- and particle-phase HOMs from α-pinene ozonolysis were systematically studied in a separate work (Zhao et al., 2023). These studies suggested potential particle-phase reactions to explain the discrepancies between the two phases. Unfortunately, the effects of the thermal desorption process on HOMs detection (e.g. temperature-dependent sensitivity of different HOM species) and vapor losses within the VIA were unclear back then, limiting a quantitative investigation of the particle-phase HOMs in SOA. Therefore, a thorough characterization of the entire system is needed to better understand the mass spectra it provides.

In this work, we performed an extensive characterization of the VIA–NO$_3$-CIMS system to quantify the concentrations of particle-phase HOMs, formed in the α-pinene (C$_{10}$H$_{16}$) ozonolysis system. In order to minimize the vapor losses of those (extremely) low-volatility HOM species after the VIA, a dedicated sheath flow unit (as shown in Figure 1, part 3) was designed as the main hardware update from the initial version of the VIA (Häkkinen et al., 2023). Then, we evaluated the performance of the different components of the VIA setup and the interface to the NO$_3$-CIMS, and further characterized how the signals changed as a function of variations in different parameters of the VIA and the sample, including evaporation temperature, flow rates, and particle sizes. In addition, by scanning the VIA temperature, we investigated that volatility information can be inferred from the measured thermograms. The homogeneous series of polyethylene glycol (PEGs) was used to evaluate the volatility measurements. Finally, we constructed a simple one-dimensional model to estimate the concentration of the sampled particles by accounting for vapor losses within the VIA vaporization tube.



## 2 Instrumentation and experiments

### 2.1 Vaporization Inlet for Aerosols (VIA)

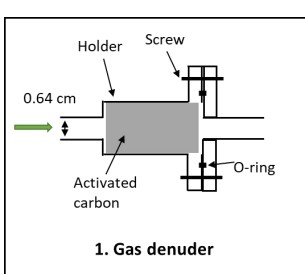 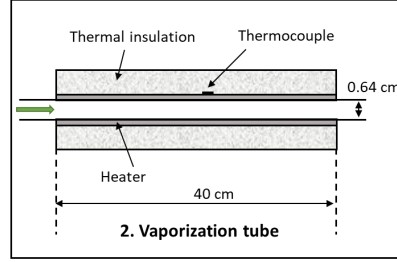 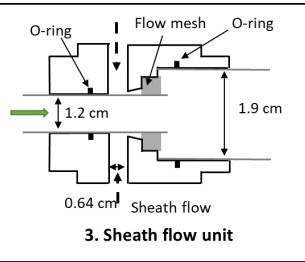

**Figure 1. A schematic diagram of the new VIA inlet includes, 1) a gas phase denuder (length: ~10 cm), 2) a vaporization tube (length: ~40 cm), and 3) a sheath flow unit (length: ~5 cm). The green arrows indicate the direction of the sampling flow, and the figures are not drawn to scale with their physical dimensions.**

The VIA (Vaporization Inlet for Aerosols, Aerodyne Research, Inc.) was designed to provide online measurements of compounds in the particle phase and to be coupled with different types of gas monitors for analysis, e.g. chemical ionization mass spectrometers. Its current commercial version contains the first two parts in Figure 1, i.e. an activated carbon gas denuder to remove the gas compounds and a vaporization tube for particle evaporation, and the coupling of this VIA with a $NO_3$-CIMS has been described by Häkkinen et al. (2023). The sheath flow unit was designed in this work to minimize the wall loss of the evaporated hot vapors.

### 2.1.1 Gas phase denuder

A honeycomb-channeled activated carbon gas denuder (Fig. 1) is deployed to remove gas compounds while transmitting particles. The cylindrical gas denuder (length: 4 cm, outer diameter: 2 cm) is mounted in a plated aluminum holder, the two halves of which are sealed with a Viton O-ring, and have external ¼-inch tube ends (i.e. outer diameter: 0.64 cm). The activated carbon denuder can be regenerated by flushing through clean air of ~100 °C for at least 4 hours.

The gas-phase removal efficiency was evaluated with a certified gas-phase mixture of 13 volatile organic compounds (VOCs) diluted into nitrogen gas in a cylinder (Apel-Riemer Environmental Inc., USA), containing Acetaldehyde, Acetone, Isoprene, Methyl Vinyl Ketone (MVK), Methyl Ethyl Ketone (MEK), Benzene, Toluene, Hexanal, m-Xylene, p-Xylene, α-pinene, 1,3,4-Trimethylbenzene, and Naphthalene (detailed information in Table S2). The particle transmission efficiency of the gas denuder was evaluated by using monodispersed ammonium sulfate (AS, Sigma-Aldrich) particles (20-700 nm) generated from an atomizer (Aerosolgenerator ATM 220, TOPAS, Germany). The results of the above tests will be discussed in section 3.1.1.

### 2.1.2 Vaporization tube

A vaporization tube (length: 40 cm, outer diameter: 0.64 cm), made of Sulfinert coated stainless steel (i.e. bonding an inert silica layer into the surface) and covered by a 24-V resistive heating element and glass-wool insulation, is used to initiate the evaporation of particles (Fig. 1). A thermocouple attached to the surface of this vaporization tube was used to monitor the temperature. The heating temperature can be manually set to a fixed value (25-350 °C) or connected to a Vocus instrument (Tofwerk AG) or an Eyeon box (Aerodyne Research, Inc.) to be programmed either in the ramping mode (i.e. continuously heating up within minutes to hours) or in the steps mode (i.e. jumping among several temperature stages). The Eyeon software uses a proportional-integral-derivative controller based on the monitored temperature to adjust the



voltage output applied to the heating element to closely follow the setting values. In addition, volatility information could
       be estimated based on the ramping mode dataset (i.e. thermograms) and will be discussed in section 3.2.

       The gas and particle transmission efficiency of the vaporization tube at different temperatures (25, 100, 200, and 300 °C)
       was evaluated by using the same VOCs cylinder described in section 2.1.1 and monodispersed sodium chloride (NaCl,
       Sigma-Aldrich, 20-700 nm) particles generated from an atomizer, respectively. The results will be discussed in section
160    3.1.2.

### 2.1.3 Sheath flow unit

       A sheath flow unit (Fig. 1) was designed in this work to cool down the sampling flow and minimize the wall loss of the
       evaporated vapors after the vaporization tube. Meanwhile, the sheath flow is also necessary to compensate for instrument
       configurations that need a large inlet flow (e.g. atmospheric pressure source such as an Eisele-type $NO_3$-CIMS (Eisele and
Tanner, 1993)). Previously, a cross-fitting unit was used for the connection, and large vapor losses were inferred from the
       concentration comparisons between the measured HOMs vapor and evaporated particles (Häkkinen et al., 2023; Zhao et
       al., 2023). In this newly designed piece, zero air or pure $N_2$ can be used as a sheath flow, which merges with the central
       sampling flow after a dense stainless-steel mesh (to keep the sheath flow as laminar as possible). Nevertheless, tubing
       length before and after the sheath flow unit, and the ratio of flow rates between the sampling and sheath line through this
unit were found to be two critical factors that affect the final sensitivity of this entire system. The results will be discussed
       in section 3.1.3. Note that the current version of this sheath flow unit, with a ¾-inch (i.e. outer diameter: 1.9 cm) output,
       was designed for the Eisele-type nitrate CI inlet (Eisele and Tanner, 1993).

### 2.2 Other instrumentation

       In this section, the instruments used for the characterization of the VIA are described in detail. A $NO_3$-CIMS (Tofwerk
AG/Aerodyne Research, Inc.) was coupled with the VIA to detect particle-phase HOMs with high selectivity and sensitivity
       (Jokinen et al., 2012; Ehn et al., 2014). The sampling flow is 10 L min$^{-1}$ along with 20 L min$^{-1}$ of sheath flow to minimize
       the wall loss of HOMs within the CI inlet. A soft X-ray source was used to ionize $HNO_3$ to $NO_3^-$ ions, which were directed
       from the sheath flow into the sampling flow by an electric field. The $NO_3$-CIMS was equipped with a long time-of-flight
       mass spectrometer, providing a mass resolution of ~8500 above 125 Th. The calibration was conducted using sulfuric acid
(SA) formed by the oxidation of $SO_2$ by OH (Kurten et al., 2012), and a calibration factor of $4\times10^9$ cm$^{-3}$ (± 50%) was
       obtained to convert the raw signals of HOMs (normalized by the sum of reagent ions) to concentrations.

       A proton transfer reaction time-of-flight mass spectrometer (PTR-TOF 8000, Ionicon Analytik Gmbh) along with the
       calibration cylinder of 13 VOC standards (described in section 2.1.1) was used to evaluate the removal efficiency of the
       gas phase denuder and the transmission efficiency of the vaporization tube. A detailed description of the PTR-TOF was
given by Jordan et al. (2009). The inlet flow is 1 L min$^{-1}$ with 0.1 L min$^{-1}$ being subsampled into the ion drift tube, which
       was operated at a pressure of ~2.6 mbar, a temperature of 60 °C, and a voltage of 600 V (with reduced electrical field
       strength parameter E/N of 115 Td). Based on these settings, the primary ion isotope $H_3^{18}O^+$ (at 21 Th) was 5800 counts per
       second (cps) and the mass resolution at 137 Th ($C_{10}H_{17}^+$) was ~4500.

       A long Time-of-Flight Aerosol Mass Spectrometer (LTOF-AMS, Aerodyne Research, Inc.) was used to measure the mass
concentration of SOA, with a mass resolution of ~8000. Using an aerodynamic lens to focus particles and thermal
       evaporation (600 °C) followed by electron impact ionization (70 eV), the mass concentration of submicron OA and
       inorganic particles can be obtained (Decarlo et al., 2006; Canagaratna et al., 2007). The ionization efficiency of AMS was
       calibrated by using 300 nm ammonium nitrate particles, and the default relative ionization efficiency of 1.4 was used for
       organics. Note that the above mass spectra datasets were analyzed either by the MATLAB-based tofTool (version 607)





package or the Igor-based Tofware (Tofware_v3_2_3) and ToFAMS (ToF_AMS_HRAnalysis_v1_25A) package. In addition, a custom-made Vienna-type differential mobility analyzer (DMA, 10-800 nm) (Reischl et al., 1997) and a commercial condensation particle counter (CPC 3750, TSI) were used to measure the size distribution of aerosol particles. The inner and outer radii of the DMA electrodes are 2.5 and 3.3 cm, respectively, with the effective electrode length being 28 cm.

**2.3 Experiments**


In order to characterize the performance of this VIA–NO$_3$-CIMS system, we used several stable particle sources as the input (summarized in Table S3). First, a single-component solution of AS was used to generate monodispersed aerosol particles. Different sizes and number concentrations of AS particles were used to evaluate the evaporation and detection efficiency of the system. Next, a solution of polyethylene glycol mixtures (PEG 400, a mixture of different PEG oligomers with an average molar mass between 380-420 g mol$^{-1}$) was used to evaluate the volatility measurements. The homogeneous

series of PEGs was recommended and used by previous studies as benchmark molecules for volatility measurements owing to their chemical and thermal stability (Krieger et al., 2018; Bannan et al., 2019). PEGs are in the liquid phase at room and measurement temperatures, and thus can be easily prepared and used to mimic the liquid or amorphous solid OA particles (Cappa et al., 2008). On the other hand, the intra-consistency of different organic compounds with a large variety of volatility ranges using current techniques is usually much better than inter-comparisons of the same compounds among

different techniques (Bilde et al., 2015). Thus, PEGs can be used to evaluate the systematic bias related to different instruments and systems. In addition, a potential aerosol mass (PAM) oxidation flow reactor (~13 liters, stainless steel) was used to generate a multi-component organic mixture, with SOA mass concentrations ranging from 7.23 to 94.4 µg m$^{-3}$, from α-pinene ozonolysis under dry conditions. The total flow rate through the PAM reactor was 8 L min$^{-1}$, resulting in

a residence of ~1.6 min. For a more detailed concept and description of the PAM, please refer to previous works (Kang et al., 2007; Lambe et al., 2011). Note that the identified HOM species, with six or more oxygen numbers from the α-pinene ozonolysis reactions (Bianchi et al., 2019), were grouped into different carbon number families as described in Zhao et al. (2023). HOM monomer and HOM dimer (i.e. dimetric accretion products) refer to C$_8$-C$_{10}$ and C$_{16}$-C$_{20}$ compounds, respectively, while the peaks of other carbon numbers observed are named as HOM others in the following discussion.

**2.4 Description of the fitting method for the thermograms**

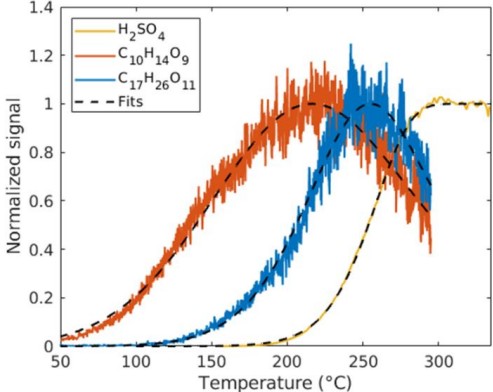

**Figure 2: Measured (solid lines, 10-second dataset) and fitted (dashed lines) thermograms for sulfuric acid (SA, H$_2$SO$_4$), C$_{10}$H$_{14}$O$_9$, and C$_{17}$H$_{26}$O$_{11}$.**



Thermograms can be obtained during temperature-programmed thermal desorption. The observed vapor concentration of

a given compound is determined by its evaporation from the particles and losses of the evaporated vapors. In the FIGAERO design, a limited amount of particles are collected on the Teflon filter and a maximum of the vapor signal is observed after the full evaporation of that compound (Lopez-Hilfiker et al., 2014). The temperature corresponding to that maximum in the thermogram ($T_{max}$) is related to its volatility (Schobesberger et al., 2018; Thornton et al., 2020). In the VIA, real-time evaporation of particles is achieved without pre-collection. Assuming negligible losses of vapors within the vaporization

tube, we expect that the evaporated fraction of aerosol particles will increase as the VIA temperature rises and reach a plateau (instead of a single maximum) after full evaporation. The plateau is expected since the vaporization tube is constantly supplied with new particles in the sampling flow through the VIA. Therefore, there will be no decrease in the signal due to running out of mass to evaporate. This sigmoid-shape of thermogram was obtained when using ammonium sulfate as the particle source in the experiments, during which sulfuric acid was generated and measured using the VIA–

NO$_3$-CIMS system (Fig. 2).

In order to simulate the thermogram, the temperature-dependent rate of evaporation of compound $X$ from the particle phase is adapted from the one used by Schobesberger et al. (2018):

$$\frac{d[X]_{particle}}{dt} = -[X]_{particle} \sqrt{\frac{T^*}{T(t)}} \cdot \exp\left(-k\left(\frac{1}{T(t)} - \frac{1}{T^*}\right)\right) \tag{1}$$

where $T^*$ and $k$ are the free parameters to be fitted based on the measurements. The temperature inside the vaporization

tube as a function of time $T(t)$ is obtainable by combining the temperature profile inside the tube (Fig. S1) with the flow rate as a function of temperature. A more detailed description of Eq. (1) is given in section S1 in the supplement. Then, the evaporated concentration of compound $[X]_{evaporated}$ can be described by:

$$[X]_{evaporated} = [X]_0 - [X]_{particle} = [X]_0 - [X]_0 \exp\left(\int_{t=0}^{\tau} -\sqrt{\frac{T^*}{T(t)}} \cdot \exp\left(-k\left(\frac{1}{T(t)} - \frac{1}{T^*}\right)\right) dt\right) \tag{2}$$

where $[X]_0$ is the initial mass concentration of compound $X$ in the particle phase, which also needs to be fitted, and $\tau$ is the

total residence time in the vaporization tube. The fitting of SA signals using this model shows good agreement with the SA measurements (Fig. 2). In contrast to the SA experiments, HOM molecules (multi-functional organic compounds evaporated from SOA particles) may be lost upon impacting the hot walls of the vaporization tube. Indeed, thermograms which look like those obtained by the FIGAERO system were also measured for HOM compounds using the VIA–NO$_3$-CIMS system. We assume this is owing to the vapor losses within the vaporization tube, with earlier evaporation in the

tube causing a larger fraction of a compound to be lost, thus reaching a single maximum (at $T_{max}$) during the thermal desorption ramp (examples shown as orange and blue lines in Fig. 2). It is important to note that the mechanism behind the shape and peak of the VIA thermogram is entirely different from the FIGAERO systems. For any organic compound $X$, the rate of the impacts is assumed to be proportional to its diffusion coefficient $D_X$ and gas-phase concentration $[X]_{gas}$:

$$\frac{d[X]_{gas}}{dt} = -\frac{d[X]_{particle}}{dt} - cD_X(T)[X]_{gas} \tag{3}$$

where a proportionality constant $c$ is included in the model. $c$ is assumed to be instrument specific and constant for different temperatures and compounds, and it was determined by manually checking the thermograms for several different compounds (Fig. S2). Note that a manual approach for determining $c$ was preferred to ensure the quality of the thermograms used. $D_X(T)$ was obtained using the Fuller method (Fuller et al., 1966; Tang et al., 2014), which has previously been used for determining the diffusion coefficients of HOMs (Peräkylä et al., 2020). The free parameters ($[X]_0$, $T^*$, and $k$) were

determined by a least-square fit of the numeric solution of $[X]_{gas}$ from Eq. (3) to thermogram data (dashed lines in Fig. 2). The corrected signal (i.e. without vapor losses) is then given by the fitted value of $[X]_0$. This model for correcting the signal is one-dimensional and does not account for any radial variations in temperature and flow rate. Our model only includes a "bulk" aerosol mass, i.e. it does not account for different particle sizes or any kinetic limitations that may exist in the





particles. Effects of thermal decomposition of molecules in advance of interaction with the walls are not included. A more
detailed explanation of the fitting results for the thermogram of sulfuric acid and HOM species will be given in Section 3.3.1.

## 3 Results & Discussion

### 3.1 Characterization of the VIA inlet

#### 3.1.1 The gas phase denuder

Gas removal efficiency and particle transmission efficiency are the two key parameters to evaluate the performance of a gas denuder. Based on the PTR-TOF measurements (Fig. S3a and Fig. S4) with a flow rate of 1 L min$^{-1}$, removal efficiencies of 97.2-99.9% were observed for 13 different VOCs (~8 ppb). In addition, ~80% of 500-800 ppb $\alpha$-pinene and >95% of 4 ppm $O_3$ were removed (with a 1.5 L min$^{-1}$ flow rate) during one test within the PAM chamber. Based on the CPC measurements of monodispersed AS particles (20-700 nm, Fig. S3b), particle losses within the gas denuder are less than
5% for particles above 50 nm. Overall, this honeycomb-activated carbon gas denuder is expected to work effectively under various lab and field conditions.

#### 3.1.2 The vaporization tube

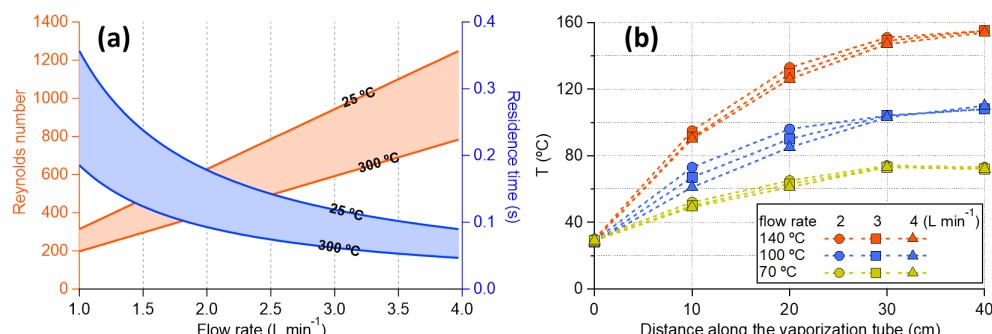

Figure 3. (a) Residence time and Reynolds number as a function of volumetric flow rate (at the input side of the vaporization
tube). (b) Temperature profile within the vaporization tube monitored with different flow rates. The temperatures were measured by inserting a thermocouple in the ¼-inch vaporization tube, and the comparable temperature profiles at different flow rates might be partly explained by the thermocouple touching the tubing wall.

Residence time, as a function of flow rates for a given tubing dimension, within the vaporization tube is a crucial parameter for its design (Burtscher et al., 2001). A longer residence means better evaporation but may result in more particle losses
(including wall loss and potential thermal decomposition), while a shorter residence time may lead to incomplete vaporization. For example, it has been found that the temperature needed to fully evaporate AS particles increase with the decrease in residence time and the increase in particle size, which may lead to some inconsistencies among different volatility measurements (Burtscher et al., 2001; An et al., 2007; Faulhaber et al., 2009).

The residence time within the VIA vaporization tube is estimated to be within a range of 0.045-0.36 s with working flow
rates of 1-4 L min$^{-1}$ and temperatures used during the experiments (Fig. 3a). Since only ~40% of the vaporization tube could reach the set temperatures (Fig. 3b), the effective residence time (for evaporation) would be roughly half of the above estimations. The residence time is much shorter than those (9-63 s) used in previous thermodenuder designs (Burtscher et al., 2001; Wehner et al., 2002; An et al., 2007; Huffman et al., 2008), which focused on measuring the non-volatile part of the particles at different temperatures. Consequently, much higher temperatures (~300 °C, Fig. 4a) are needed to fully





evaporate AS particles than those reported by previous TD studies, e.g. 150-180 °C for 100 nm AS particles (Burtscher et al., 2001; An et al., 2007).

Assuming that the flow within the VIA vaporization tube is laminar (Reynolds number < 1300, Fig. 3a), particle losses mainly include diffusion, sedimentation, and thermophoresis onto the tube walls. All these processes are particle size, temperature, and residence time dependent. In the following discussion, we will evaluate the particle loss with a flow rate

of 1 L min$^{-1}$ as a lower limit of the particle transmission efficiency, because higher flow rates (with smaller residence time) will generally decrease particle losses. First, sedimental loss increases with the increase in particle size. For a 1-um spherical NaCl particle at 25 °C as the worst scenario, the settling velocity is calculated to be $7.74 \times 10^{-3}$ cm s$^{-1}$. With a residence time of 0.36 s, the settling distance is $2.8 \times 10^{-3}$ cm, which is much smaller than the inner diameter of the vaporization tubing (by a factor of ~155). Thus, the effect of sedimentation is negligible. Second, diffusive loss increases

with the decrease in particle size. For a 10-nm spherical NaCl particle at 300 °C as the worst scenario, the diffusion constant D is about $6.7 \times 10^{-4}$ cm$^2$ s$^{-1}$, and the root-mean-square displacement by diffusion Y is calculated to be $2.2 \times 10^{-2}$ cm, which is much smaller than the inner diameter of the tube (by a factor of ~20). Thus, the effect of particle diffusion should also be negligible. Last, thermophoresis will force the particles to the centerline, which partly compensates for the diffusion and sedimentation (Villani et al., 2007). Overall, the above rough estimations give an upper limit of the particle loss within the

vaporization tube. The detailed calculations are given in Section S2 of the supplementary.

Comparable to our above estimation, the CPC measurements (Fig. S5b) of monodispersed NaCl particles (20-700 nm) showed particle losses < 10% with a slight increase as the increase in VIA temperature. Similar results were reported from previous TD studies (Wehner et al., 2002; Huffman et al., 2008). The lowest particle transmission efficiency (89.5%) was observed for 300 nm NaCl particles at 300 °C. Thus, the particle loss through processes other than thermal evaporation will

not affect the results largely. On the other hand, near unity transmission efficiency for most of the VOC standards was observed based on the PTR-TOF measurements (Fig. S4 and Fig. S5a), indicating that this VIA system can perform well if mounted in front of a PTR-TOF to measure less oxidized organic species. The increase of some VOC species at high temperatures was mainly owing to the evaporation of those gases, condensed at low-temperature stages, from the tubing wall. Differently, the losses of more oxidized organic species (e.g. HOMs) within this vaporization tube were observed and

will be discussed in Sec. 3.2.

### 3.1.3 The sheath flow unit

After thermal desorption, vapor transmission is the key parameter that determines the sensitivity of the entire system. Within the sheath flow unit, a sheath flow was supplied to cool down the sampling flow and minimize the wall loss of the hot vapors during the transport for analysis. In addition, the sheath flow is needed to compensate for the 10 L min$^{-1}$ inlet

flow of the NO$_3$-CIMS. Compared to the evaporated organic/inorganic vapors, the diffusion of air is faster, which results in a rapid temperature drop between the vaporization tube and the sheath flow unit. Consequently, turbulence (along with thermophoretic losses) and recondensation of hot vapors may take place in the cooling area (Fierz et al., 2007). Thus, the sheath flow unit needs to be supplied immediately as the evaporated vapors exit the vaporization tube to minimize their wall losses.

However, we found that directly connecting the VIA vaporization tube to the sheath flow unit will lead to a significant drop in the total ion counts of the NO$_3$-CIMS, possibly owing to the formation of turbulence, after mixing the hot sampling flow with the cooler sheath flow, in the CI inlet tubing. Using a piece of stainless-steel tubing (length: 5, 15, or 30 cm, outer diameter: 1.2 cm, as a cooling tube before the sheath flow unit) to cool the sampling flow down before mixing with the sheath flow could help to reduce the turbulence. Alternatively, using longer inlet tubing (75 cm vs. 40 cm, after the

sheath flow unit) for the NO$_3$-CIMS, with the shortest cooling tube (5 cm), also helped to decrease the effects of turbulence.




As shown in Figure S6, this setup (purple markers) gave the best sensitivity among different setups and showed consistency of sulfuric acid mass concentrations between the SMPS and $NO_3$-CIMS measurements, indicating that the nucleation/recondensation of the evaporated vapors might not be an issue. Furthermore, comparable thermograms were observed between the SMPS and $NO_3$-CIMS measurements (Fig. 4b), confirming the negligible losses of sulfuric acid after

evaporation. Nevertheless, a slightly delay of the evaporation was observed with the increases in particle sizes (Fig. 4a). Multiple charged particles might play a role in this delay (based on our quick tests), but nearly full evaporation of 300 nm AS particles at 300 °C is already good enough for most lab and field experiments, thus we did not put much efforts to dig into the details.

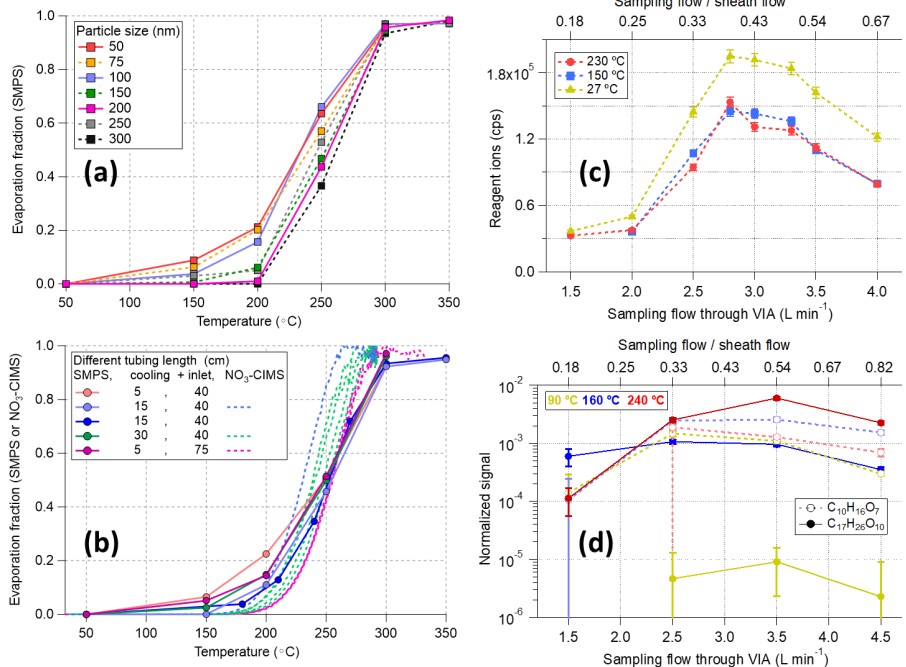

**Figure 4. Evaluation of the detection efficiency of the VIA for (a) size-selected (50-300 nm) ammonium sulfate particles with a fixed setup (i.e. 5 cm cooling and 40 cm inlet tube), and (b) different setups with 100 nm ammonium sulfate particles. The SMPS measurements in panels (a) and (b) were corrected for the size-dependent transmission efficiency (details in Figure S7). The effects of different sampling flow rates on the sensitivity of the system with (c) zero air and (d) stable SOA ($14 \pm 1.5$ µg m⁻³) input at three different temperatures. HOM signals were normalized to the reagent ions and the dilution factors of different flow ratios**
**were applied for HOM signals for more straight comparisons in panel (d) and in Figure S8.**

In addition to the length of the cooling and inlet tubing (i.e. the temperature effects on mixing flows), the flow ratios between the sampling flow (1.5-4 L min⁻¹) and the sheath flow (8.5-6 L min⁻¹), which together make up the total inlet flow of 10 L min⁻¹, could also affect the sensitivity of the VIA–$NO_3$-CIMS system. During the zero-air test, the temperature effect is not significant owing to the usage of a 15-cm cooling tube (yellow markers vs. red and blue markers in Fig. 4c).

Thus, the decrease of total reagent ion signals (i.e. the sum of nitrate monomer, dimer, and trimer) is mainly owing to the turbulence caused by mixing flows of different flow rates, altering the total ion counts by a factor of 5-6, despite the usage of flow meshes to keep the sheath flow as laminar as possible. This result highlights a well-known issue in the CIMS community that the Eisele-type nitrate CI inlet is super sensitive to flow arrangements.

The optimal sampling flow rate was observed around 2.5-3.5 L min⁻¹ for the highest total ion counts and similar trends
were observed for particle-phase HOMs measurements (Fig. 4d and Fig. S8). For example, although the $C_{17}H_{26}O_8$ signal increased significantly with temperature because of enhanced evaporation, the relative changes in signals for different flow





rates were similar for all temperatures. Furthermore, these relative changes among different HOM species (i.e. the distribution of measured HOM species) started to stabilize when the sampling flow through the VIA exceeded 2.5 L min$^{-1}$ (Fig. S9). The reason for an optimal flow ratio of around 3-to-7 is related to the current physical dimensions of this sheath

flow piece, which were designed to minimize turbulence after flow mixing, with a face velocity ratio of 1.25 between the sample and sheath flow. Note that a sampling flow of 3 L min$^{-1}$ was used in the experiments discussed in the following sections.

### 3.2 Thermogram

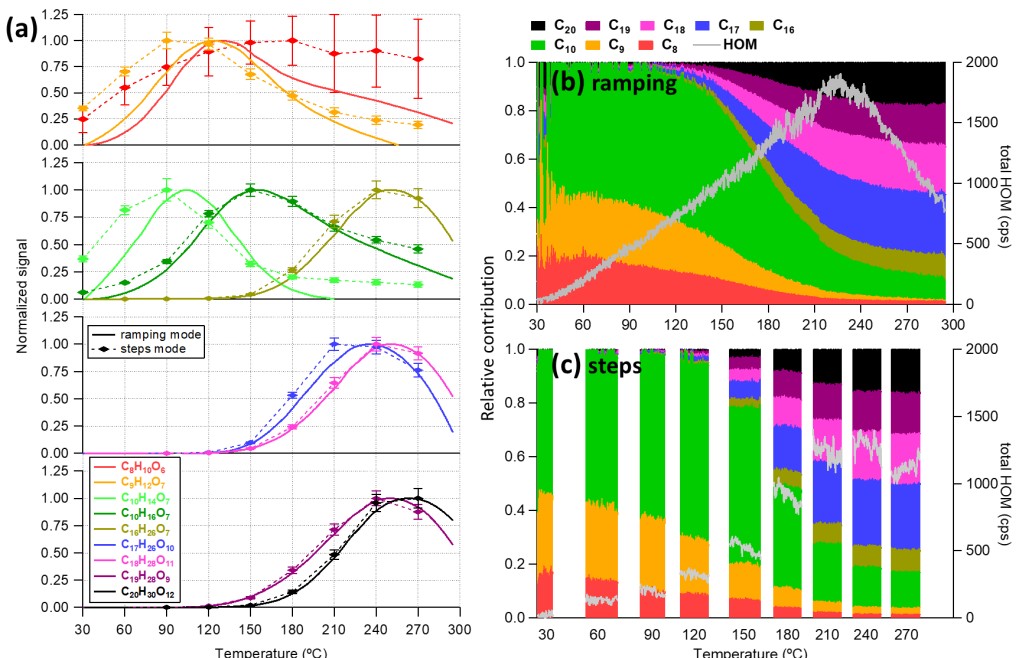

**Figure 5. Comparison of measurements obtained between the ramping and the steps mode for (a) thermogram of some chosen HOM monomers and dimers, and (b, c) the relative contributions of different C number families to the total HOM signals. In (a), smoothed signals are shown for the ramping mode ("Loess" algorithm with a bandwidth of 0.25 and the second order local polynomial was used), while the mean (diamond markers) and standard deviation (bottom and top whiskers) are shown for the steps mode. The thermograms are normalized to the reagent ions first and then to their maximums. The raw signals of the same**

**dataset are given in Fig. S10.**

In this section, we investigated the performance of the VIA–NO$_3$-CIMS system when sampling from a stable aerosol source (a PAM oxidation flow reactor for SOA particles or an atomizer for size-selected particles). The VIA used in this work can be operated by ramping the desorption temperature from 25 to 350 °C in the ramping mode or setting the temperature to several temperature steps of interest in the steps mode. The particle-phase HOM signals measured between these two modes

are compared in Figure 5, where the thermogram of some HOM monomers (C$_8$-C$_{10}$ with $T_{max}$ ~100-150 °C) and dimers (C$_{16}$-C$_{20}$ with $T_{max}$ >200 °C) are shown. In general, both the thermograms of most low-volatile HOM species (Fig. 5a) and the distribution of bulk HOM compounds (Fig. 5b, c) between these two modes are quite comparable, although the total HOM signals obtained in the steps mode are ~18% lower than in the ramping mode. The lower HOMs concentrations observed in the steps mode can be partly explained by lower SOA mass concentrations compared to the ramping mode

measurements (Fig. S10). In addition, the largest difference is on the thermograms of relatively high-volatile compounds (e.g. C$_8$H$_{10}$O$_6$ and C$_9$H$_{12}$O$_7$), which are less selective by the NO$_3$-CIMS (Hyttinen et al., 2015).



**3.2.1 Volatility**

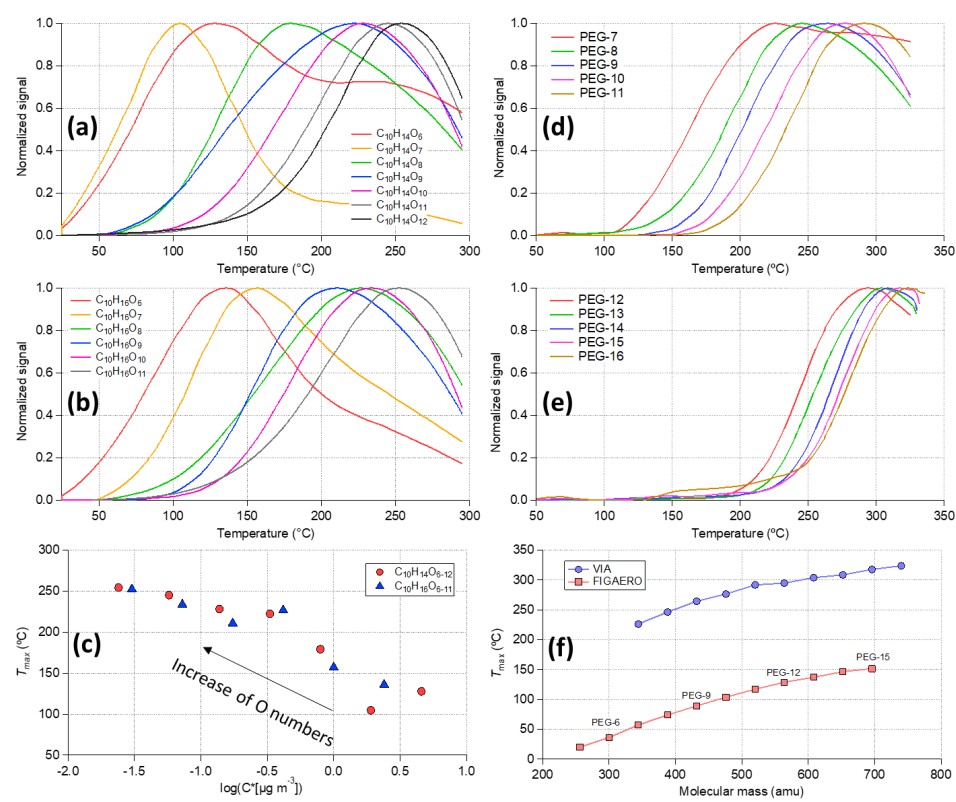

**Figure 6. Thermogram of (a) $C_{10}H_{14}O_{6-12}$, (b) $C_{10}H_{16}O_{6-11}$, and (d, e) polyethylene glycols (PEGs). (c) The scattering plot of $T_{max}$ obtained from panels (a) and (b) vs. the estimated saturation vapor concentration using the composition-determined empirical expression in Peräkylä et al. (2020). (f) $T_{max}$ as a function of molecular mass (atomic mass unit) for PEGs measured by the VIA–NO₃-CIMS and the FIGAERO-iodide-CIMS (Ylisirniö et al., 2021).**

Figure 5a shows a very interesting trend of decreasing HOM signals after reaching their maximums as described in Sec. 2.4. This decrease at high temperatures can be largely explained by vapor losses after evaporation within the vaporization tube, but this is not the case for sulfuric acid and VOCs as discussed in section 2.4 and 3.1.2, respectively. In particular, for each HOM compound, lower temperatures are not enough to fully evaporate the particle phase, while higher temperatures might cause earlier evaporation within the vaporization tube thus with larger losses (i.e. HOM vapors collide with and loss to the walls). Therefore, the maximum signal (and corresponding $T_{max}$) in the measured thermogram is determined by the balance between the evaporation rate and the loss rate.

In Figure 5a, HOM compounds containing more carbon and oxygen atoms tend to show higher $T_{max}$ (i.e. lower volatility), and this trend conserves within the $C_{10}H_{14}O_z$ and $C_{10}H_{16}O_z$ families (Fig. 6a, b) as well. The measured $T_{max}$ of these compounds is related to their saturation vapor concentrations (Fig. 6c) estimated using the composition-determined empirical expression given in Peräkylä et al. (2020), indicating that $T_{max}$ can be used to represent the volatility of measured organic compounds as used in the FIGAERO measurements (Lopez-Hilfiker et al., 2014). Next, the homogeneous series of PEGs was used (Fig. 6d-6f) to compare the volatility measurements between the VIA–NO₃-CIMS (identified PEG peaks are summarized in Table S4) and the FIGAERO-iodide-CIMS (Ylisirniö et al., 2021) systems. Similar to the ammonium sulfate experiments discussed in Section 3.1.2, using milli-second residence time within the VIA heating tube, much higher $T_{max}$ was obtained compared to other thermal desorption techniques (e.g. the FIGAERO and TD systems). In the FIGAERO,

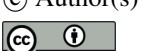



the collected particles are continuously heated up (10-50 K min$^{-1}$) on the filter. For the VIA, the time spent above a given

temperature is an appropriate factor that can be used for determining the evaporation (Fig. S11). Furthermore, the very short residence time of the VIA can explain the shift of the $T_{max}$ to higher values for PEGs (Fig. 6f) and HOMs (Fig. 7a) compared to previous FIGAERO measurements (Ylisirniö et al., 2020; Ylisirniö et al., 2021). Nevertheless, a near-perfect-linear relationship of measured $T_{max}$ for the PEGs was found between these two systems (Fig. S12), indicating the potential to infer volatility from the measured thermograms using the VIA–NO$_3$-CIMS system.

Another very interesting feature of the thermograms is that most of the HOMs, including monomers, only showed a single mode. Comparable or even slightly higher contributions of HOMs dimers than monomers were observed in this (Fig. 9) and previous VIA works (Häkkinen et al., 2023; Zhao et al., 2023). These results suggest that thermal decomposition products are rarely detected by the VIA–NO$_3$-CIMS, unlike what has been reported in the FIGAERO setup (Lopez-Hilfiker et al., 2015). Some exceptions exist, e.g. the signal of $C_{10}H_{14}O_6$ (Fig. 6a), which also seems to have a contribution from

decomposition of larger molecules. Nevertheless, the combination of the VIA and the NO$_3$-CIMS has the potential to detect also weakly bonded peroxide (ROOR) accretion products (e.g. $C_{20}$ dimers in α-pinene SOA). Other CIMS instruments, with selectivity towards less functionalized and more volatile species, may be able to detect more of the decomposition products.

### 3.2.2 Reproducibility of thermograms

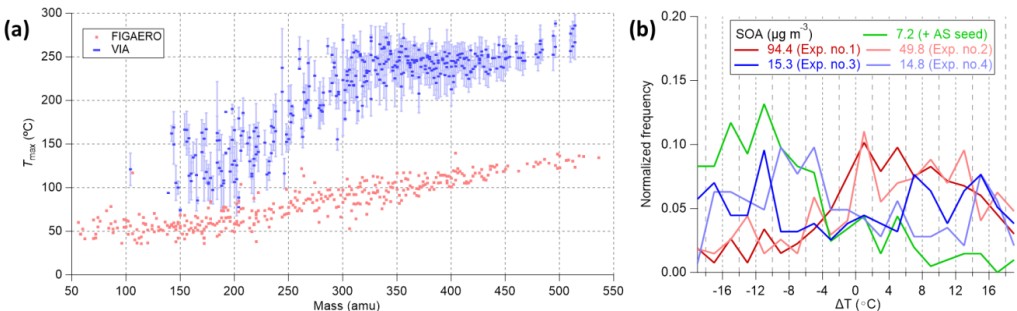


**Figure 7. (a) $T_{max}$ for α-pinene SOA measured by the VIA−NO$_3$-CIMS system (blue markers for the mean and light blue whiskers for the one standard deviation) and the FIGAERO-iodide-CIMS adapted from Ylisirniö et al. (2020). (b) The normalized distribution of $T_{max}$ difference (ΔT) between each experiment and the mean of five experiments for HOMs. In panel (b), only the 7.2 µg m$^{-3}$ SOA experiment was conducted using ammonium sulfate as the seed particles.**

It is crucial to ensure the reproducibility of the thermograms to estimate the uncertainty when relating the measured $T_{max}$ to the volatility for organic species. The calibration method (syringe deposition vs. atomization), solution/mass concentration, particle size, and heating ramp rate are the factors that reported could largely affect the volatility (i.e. $T_{max}$) in the FIGAERO-iodide-CIMS system (Ylisirniö et al., 2021). The effects of SOA mass concentration on the measured $T_{max}$ using the VIA–NO$_3$-CIMS system were investigated in this section. As shown in Fig. 7b, the $T_{max}$ of HOMs measured

under lower SOA mass concentrations showed negative bias from the mean of the five experiments, while the $T_{max}$ under higher SOA mass concentrations showed positive biases from the mean value. In addition, the $T_{max}$ measured under comparable SOA conditions correlated well (Fig. S13b). On the other hand, the delay of $T_{max}$ with the decreases in the residence time and the increases in particle sizes were discussed in Sec. 3.2.1 and in Sec. 3.1.3, respectively. Furthermore, when the 100-nm AS particles were coated with organics, the evaporation of sulfuric acid was delayed (Fig. S13c), while

the temperature ramping rate did not affect the evaporation because the thermal desorption process is different from the FIGAERO system. Overall, most of the factors that affect the volatility measurements in the previous FIGAERO and TD




systems seem also played an important role in the VIA, and these effects warranted a more systematic investigation in future works to obtain more accurate volatility measurements using the VIA–NO$_3$-CIMS system.

### 3.3 Quantification of particle-phase HOMs

**3.3.1 Fitting the thermograms**

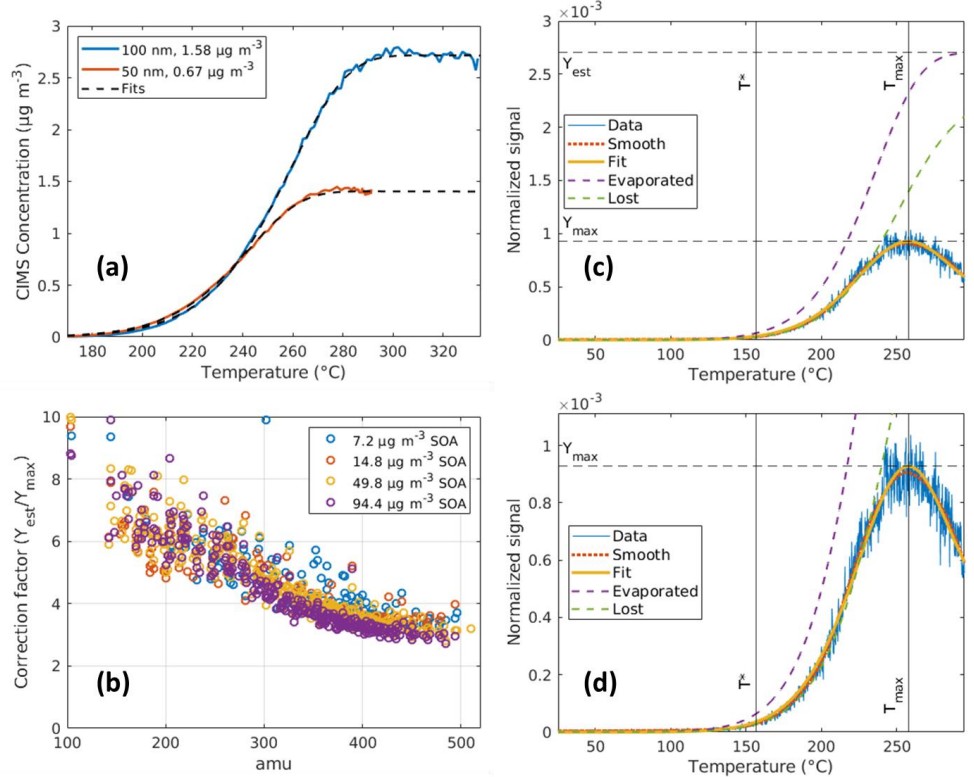

**Figure 8: Fitting results of the observed thermogram for (a) sulfuric acid and (c) C$_{20}$H$_{30}$O$_{12}$. (b) Summary of the correction factor (CF) for the VIA−NO$_3$-CIMS system in terms of particle-phase HOMs measurement, the mass dependency of these curves make sense because heavier HOMs evaporate later, thus smaller correction is needed. (d) The same result as panel (c) but zoomed in**
**on the details to show the agreement between the fit and measurement. In panel (c), the "Data" curve is the measured signal of C$_{20}$H$_{30}$O$_{12}$ normalized to the reagent ions. The "Smooth" curve used loess smoothing with a bandwidth of 0.25 on the data. The "Evaporated" curve represents the estimated signal without gas phase losses using Eq. 1, which shows the estimated maximum signal ($Y_{est}$). The "Lost" curve describes the cumulated diffusional vapor losses (assuming 100% loss once a molecule hit the wall) within the VIA vaporization tube. The "Fit" curve displays the calculated gas-phase signal as the difference between the**
**"Evaporated" and "Lost" curves. More examples of the fitting result for other HOMs and the fitted parameters ($T^*$ and $k$) are shown in Figure S14 and S15, respectively.**

In this section, we used the one-dimensional model described in Section 2.4 to simulate the evaporation of particles and the loss of evaporated HOMs vapors within the vaporization tube to fit the measured thermograms. As the temperature ramps up, the particles within the sampled aerosol start to evaporate earlier inside the vaporization tube. Without significant
losses for sulfuric acid, the measured signal increases with the temperature and reaches a plateau after full evaporation. This hypothesis is supported by the good agreements between the measurements and the fitted signal using only the evaporation function (Figure 8a). For HOMs, the situation is more complicated, and an extra loss term is added to fit the thermograms (Figure 8c, d). The competition between the evaporation and loss of HOM vapors leads to the thermogram rising and later falling as the VIA temperature ramps up. We suspect this loss is mainly owing to that HOM molecules





were lost upon impacting the walls of the vaporization tube. The earlier the particles evaporate within the vaporization tube, the longer the time HOMs vapors had to diffuse to the walls. This leads to a significant loss of signals at higher temperatures, where most of the particle mass has evaporated before reaching the end of the vaporization tube. Consistently, a mass dependency of the estimated correction factor was observed (Fig. 8b), because heavier HOMs evaporate later, then smaller correction is needed. However, lower temperatures may not be able to fully evaporate the low-volatile species.

Consequently, the measured signal shows one single maximum ($Y_{max}$), which unfortunately only represents a fraction of the total particle-phase concentration that survived to enter the NO$_3$-CIMS. Therefore, the fitting method to the thermograms is needed to obtain a more accurate estimation of the particle-phase HOMs concentration in the aerosol sample.

Here, we define the correction factor (CF) as the ratio of the estimated initial particle-phase concentration ($Y_{est}$, fitted

maximum) and the measured maximum $Y_{max}$ in thermogram. The CF obtained for several datasets (with SOA mass concentrations of 7.2 – 94.4 µg m$^{-3}$) is relatively comparable (Fig. 8b). However, the CF is quite sensitive to the proportionality constant c, which also has some uncertainties associated with it (Figure S2 and Table S5). This means that while the relative correction factors between different compounds may be well known, the absolute values can still vary by an order of magnitude. An additional source of error in the CF values is the estimated diffusion constants, these may be

off by up to 10%. There is also little information available about how the Fuller method estimated diffusion coefficients scale up to temperatures of >300 ºC (Tang et al., 2015). In addition, the lighter species show more erratic and less consistent thermograms, and this may introduce larger uncertainties compared to low-volatile larger molecules.

**3.3.2 Particle-phase HOM**

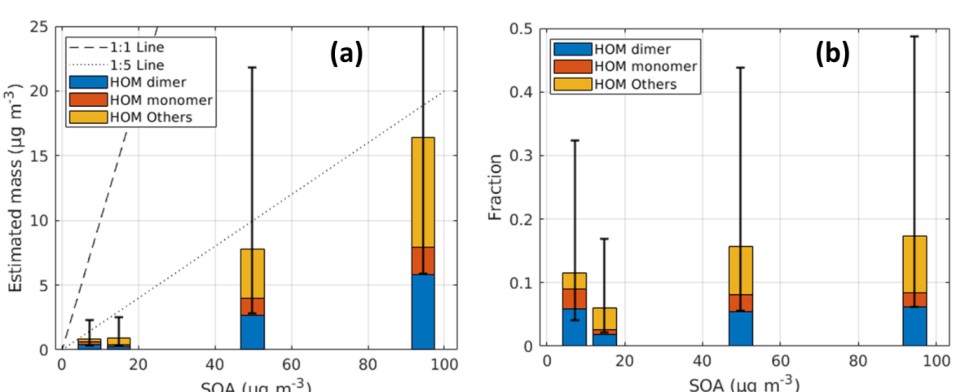

**Figure 9. Estimated (a) mass concentration and (b) mass fraction as a function of SOA mass concentration. HOM monomer, HOM dimer, and HOM others refer to C$_8$-C$_{10}$, C$_{16}$-C$_{20}$, and HOMs of other C numbers measured in the NO$_3$-CIMS mass spectra. SOA mass concentrations were measured by the AMS.**

Based on the correction factor obtained for each compound, the mass concentration of particle-phase HOM can be estimated as shown in Figure 9a. Though with large uncertainties, our best estimate for the contribution of particle-phase HOMs to

the total SOA mass is ~15%. We hope that future studies will be able to narrow down the uncertainties. A linear relationship was measured between the particle-phase HOM and SOA, with a roughly comparable distribution between HOM monomers and dimers among all experiments (i.e. dimers make up between 65-74 % to the total HOM monomers and dimers mass), suggesting that while our absolute quantification is very uncertain, the reproducibility of the measurements is much better. The linear relationship also applies to most individual species, disregarding some outliers (Fig. S16). The

contribution of the "HOM others" increases with the increase of SOA mass concentrations, which can be largely explained



by the gas-particle partitioning theory because HOM others is dominated by compounds with carbon numbers less than 8. The estimated fractions of particle-phase HOM monomers and dimers are expected to have smaller uncertainties than the estimated mass, since the relative uncertainties of CF for different species are smaller than for their absolute values.

### 3.4 Current challenges and future improvements

In the current design of the VIA–NO$_3$-CIMS system, the time resolution of particle-phase HOM measurements (i.e. one entire temperature ramping up and down cycle) is limited by the lack of a cooling system. Under ambient conditions, at least 20-30 minutes are needed for its natural cooling down to room temperature. In future works, actively cooling the vaporization tube can be considered. If the cooling rate is larger than the heating rate, one could control the cooling rate by controlling the heat supply to make the temperature decrease linearly. Thus, the measurements during the cooling stage

can also be useful, doubling the duty cycle (if the time for ramping up and down is the same). This is feasible since we already showed that the results of ramping up and stepping down (Fig. 5) are quite comparable for most HOM species. Alternatively, the cooling stage can be used for the gas-phase HOMs measurements. Then, a separate gas sampling line and an automatic valve would be needed to couple in the current system.

In addition, a very large correction factor (~3-9) is needed to quantitatively estimate the concentrations of the particle-

phase HOMs, suggesting significant losses within the VIA system. On the one hand, the uncertainty of this correction factor could potentially be reduced significantly if the value of correlation coefficient could be determined experimentally. For example, using some known organic aerosol as the standards, this value could be determined or calibrated more exactly. This would yield better quantitative estimates of the particle phase chemical composition. On the other hand, based on our experiments, 1) particle loss was <10% within the entire system based on the NaCl test; and 2) vapor loss was negligible

for sulfuric acid and VOCs, indicating nucleation/recondensation should be negligible and less oxidized organics did not decompose during thermal desorption, respectively. However, the thermogram of HOMs indicates significant vapor loss after evaporation. Using larger diameters of the vaporization tube and larger sampling flow rates  may help to reduce the vapor loss but would reduce the heat transfer at the same time. Then, the small working flow rates of the gas denuder would be the limiting factor. Packing several gas denuders in parallel can be one solution to run the system with larger flows while

effectively removing the gaseous compounds. In that case, the sheath flow unit may needs to be designed to couple with a different flow configuration.

Finally, although this work focused on the particle-phase HOMs measurements by using a NO$_3$-CIMS, the VIA could be adapted to other CI inlets or gas analyzers in general. Different types of detectors attached after the VIA could give us a more complete picture of the molecules in (SOA) particles. As mentioned in Sec. 3.1.2, negligible loss was observed for

most of the tested VOC standards, indicating that a much less wall losses would be expected if one put the VIA in front of a detector measuring less oxidized organic species in SOA particles, e.g. using an iodide CIMS or a PTR-TOF (Avery et al., 2023).

### 4 Conclusion

We have characterized the VIA–NO$_3$-CIMS and its ability to measure HOM as a function of different parameters. The used

system was the same as in previous studies (Häkkinen et al., 2023; Zhao et al., 2023), with the exception that a sheath flow unit was designed and used as an interface between the VIA and NO$_3$-CIMS to decrease the vapor losses after evaporation. First, the performance of each component of the current VIA system was characterized. 1) The honeycomb-activated carbon denuder allows particles to pass through (> 95% for AS particles larger than 50 nm) while removing gaseous compounds (> 97% for tested VOCs), effectively. 2) The vaporization tube is capable of generating reproducible temperature profiles




with working flow rates of 2-4 L min$^{-1}$ and with sub-second residence time (e.g. 0.045 s with 4 L min$^{-1}$ at 300 ºC), which is still enough to fully evaporate AS particles < 300 nm. The small residence time may benefit the detection of (hydro)peroxides under thermal desorption using this system. In addition, particle losses, other than thermal desorption, were found to be <10% with NaCl particles, while a near-unity transmission was obtained for the tested VOCs in the vaporization tube at measurement temperatures, indicating great performances would be expected if coupling the VIA with

a PTR-TOF or an iodide-CIMS.

Then, with an optimized setup to connect the sheath flow unit to the current VIA–NO$_3$-CIMS system, we were able to obtain thermograms by using temperature-programmed thermal desorption for different aerosol particles, including, AS, polyethylene glycol mixtures (PEGs), and α-pinene SOA particles. Unlike sulfuric acid, the decrease of signals after reaching the maximum in the thermograms of HOMs, indicating significant vapor losses upon contact with the hot walls

of the VIA. This loss potentially leads to fragmentation products that are not observable by the NO$_3$-CIMS and needs further investigation by coupling the VIA with a detector measuring less oxidized organic species. In general, the measured $T_{max}$ in the thermograms was ~150 ºC higher than the measurements of the FIGAERO-iodide-CIMS, and this difference can be largely explained by the extremely short residence time within the heating area. Nevertheless, the linear relationship of the $T_{max}$ between these two systems indicates the potential to infer volatility information from the thermograms measured

by the VIA–NO$_3$-CIMS. In addition, a one-dimensional model was used to simulate the evaporation and temperature-dependent wall losses of evaporated molecules within the VIA vaporization tube, allowing a quantitative estimate of the concentration of particle-phase HOMs. Overall, the coupling of the VIA and NO$_3$-CIMS can be a promising and useful system for online measurements of HOMs in SOA particles.


*Data availability.* Data are available upon request from the corresponding author.

*Supplement.* The supplement related to this article is available online.

*Author contributions.* ME and JZ designed the study. JZ, VM, YL, EH, and JYZ conducted the experiments. JZ, VM, and YL analyzed the data. VM developed the model. JEK designed and built the original VIA used in this study. JK designed the sheath flow unit. ME, HT, MC, QZ, and DR provided support for the experimental setup and analysis methods. JZ and VM prepared the paper with contributions from all co-authors.

*Competing interests.* Douglas Worsnop and Manjula Canagaratna work for Aerodyne Research, Inc., which developed and commercialized the Vaporization Inlet for Aerosols (VIA) used in this study.

*Acknowledgements.* The authors thank Jing Cai, Nina Sarnela, and Lauriane Quéléver for the calibration of the NO$_3$-CIMS, and Pekka Rantala for technical support. The authors thank Arttu Ylisirniö and Siegfried Schobesberger for discussing on
and sharing the FIGAERO measurements.

*Financial support.* This work was supported by funding from the Academy of Finland (grant nos. 317380, 320094, 325656, 345982, and 346370) and a University of Helsinki 3-year grant (75284132). Valter Mickwitz thanks Svenska Kultufonden (grant no. 190437) for financial support. Ella Häkkinen thanks the Vilho, Yrjö, and Kalle Väisälä Foundation for financial
support. Frans Graeffe acknowledges Svenska Kulturfonden (grant nos. 167344 and 177923) for financial support.

Open-access funding was provided by the Helsinki University Library.



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
