# Peer review of "Characterization of the Vaporization Inlet for Aerosols (VIA) for Online Measurements of Particulate Highly Oxygenated Organic Molecules (HOMs)"

_EGUsphere, 2023_

## Author Comment (AC2)

**Response to Reviewer #RC1**

**General comment**

The manuscript "Characterization of the Vaporization Inlet for Aerosols (VIA) for Online Measurements of Particulate Highly Oxygenated Organic Molecules (HOMs)" report a systematic test of VIA used with $NO_3$-CIMS to detect HOM, including the transmission efficiency, evaporation efficiency, quantification of particle-phase HOM as well as applicability for volatility measurement.

The authors found that transmission efficiency of particles (NaCl>50 nm) is >90%. Transmission efficiency of VOC was also high. Also the transmission loss for sulfuric acid vapors was negligible according to the evaporated AS particles measured by SMPS and sulfuric acid measured by $NO_3$-CIMS. Adding a sheath flow after VIA reduced markedly the wall loss of HOM. The signal of HOM increased with T first and then decreased, indicating the loss of HOM in VIA. Tmax correlated with Tmax obtained from FIGAERO-I-CIMS, but much higher (~100-150 ℃) than Tmax from FIGAERO. The loss efficiency of HOM obtained by a one-dimensional model was high (3-9) and correction factor depended on molecular weight.

Determination of particle-phase organic components on-line and on molecular level is critical to understand the formation, fate and impacts of organic aerosol. In this regard, this study presents a valuable attempt to evaluate and to optimize VIA combined with $NO_3$-CIMS to be used for HOM measurement, although there is a number of limitations and challenges to use VIA for the quantification of particle-phase HOM. This manuscript is generally well-written. I have a few comments for the authors to consider before its publication in AMT.

We thank the reviewer for the positive and insightful comments, and we answer the specific comments point-by-point below. The reviewer's comments are in **blue**, and our answers are in **black**.

**Specific comments**

**Comment #1:**

In this study, it was assumed that the loss of HOM was due to the collision with hot walls. What is the evidence for this assumption? It is possible that it was due to the decomposition in the air within VIA, which was not included in the model of this study as mentioned by the authors?

**Response:**

We wish to make the causality clear by saying that we had no a priori expectations or assumptions that decomposition on the walls would be a dominant loss process when starting these studies. Rather, we found that we could initially not explain the shapes of the obtained thermograms, but once we allowed for efficient decomposition on the walls, the observations made more sense.

Consider for example Fig. 5a, where $C_{10}H_{14}O_7$ has peaked and almost been completely lost at 150 ℃, but some dimers only start to show up at this temperature. We expect that the HOM, both monomers and dimers, largely consist of similar functional groups (carbonyls, alcohols, and peroxides). If decomposition was purely a function of temperature, and some monomers start to decompose around 100C, then we would not expect to see any HOM above 200 ℃. However, we still detect compounds peaking above 250 ℃, suggesting that there has been only marginal loss of these compounds before evaporation, but then the decomposition is fairly rapid. This makes us believe that the decomposition within the particles is very limited. Our observations could also be explained by fast decomposition after evaporation to the gas phase, but if rapid particle-phase decomposition was negligible, we did not feel that rapid gas-phase decomposition should be expected either. However, contact with a hot metal

surface could be expected to lead to fast decomposition, as the heat transfer to the molecule is likely to be much larger in this case.

In addition, as our model was able to capture the thermogram shape with this assumption of efficient wall decomposition, we felt that it was motivated to suggest this as the major loss pathway. But we acknowledge that there are uncertainties involved as explained in Comment #2, and therefore use terminology like in the abstract (*"Our results indicate that…"*) when discussing this topic. In order to clarify this point to the reviewer and future readers clearly, we added the above argument in line 399.

**Comment #2:**

How was the uncertainty in Fig. 9 derived? I suggest the authors to further discuss the uncertainty/limitations of correction factor, e.g. how the factors not considered in the model influence CF, as it is key to the quantification of particle-phase HOM.

**Response:**

The uncertainties include two main parts, calibration factor of the $NO_3$-CIMS (100%) and correction factor, which includes the proportionality constant c (40%), diffusion constants (10%), and other uncertainties (30%), as we discussed in Section 3.3.1 *"However, the CF is quite sensitive to the proportionality constant c, which also has some uncertainties associated with it (Figure S2 and Table S5). This means that while the relative correction factors between different compounds may be well known, the absolute values can still vary by an order of magnitude. An additional source of error in the CF values is the estimated diffusion constants, these may be off by up to 10%. There is also little information available about how the Fuller method estimated diffusion coefficients scale up to temperatures of $>300\,^{\circ}C$ (Tang et al., 2015). In addition, the lighter species show more erratic and less consistent thermograms, and this may introduce larger uncertainties compared to low-volatile larger molecules."* In the end, summing up the uncertainties mentioned above as sort of an upper limit leads to 180%, i.e. an uncertainty of a factor 2.8. Thus, the values of error bars used in Figure 9 are x*2.8 and x/2.8. We added the above discussion in line 482 in the revised manuscript.

About the details, the uncertainty from c is an estimate from Table S5, where the c values we considered reasonable (between 4e6 and 7e6) result in a maximum deviation in the CF of 40% from the value used (c = 5.5e6). The uncertainty arising from molecules decomposing in the gas phase instead of on impact with the walls is already captured in this variation in c, since it determines the loss rate of the species. There may be differences between individual compounds, but c is determined to match the thermogram shape for all ions, so for the total mass this uncertainty should be fairly well accounted for. In addition, there is the uncertainty arising from the model being one dimensional, not being able to truly capture variation in the radial direction, which together with the uncertainty in diffusion coefficients makes up the "other uncertainties". Of course, this is not quantifiable, but based on comparisons with 3 dimensional models we believe this to be reasonable.

**Comment #3:**

Moreover, how applicable is the correction factor for one compound (molecular formula)? For example, it one does not ramp up temperature, can the correction factor be used (considering that ramping up temperature largely limits the time resolution of the method)? Or it has to be used with a thermogram? Does the correction factor depend on functional groups other than molecular weight as shown in Fig. 8b?

**Response:**

The correction factor is derived from a fitting method that relies on the measured thermograms, so determining it requires a temperature ramp. The exact shape of the thermogram depends on the setup

(e.g. flow rates and tubing length), and cannot be assumed to be "universal". In addition, the way we defined the correction factor, relates the true particle concentration to the peak of the thermogram. Without a temperature ramp, one would need to assume where the thermogram for a compound's peaks. In our experience, the temperature ramping provides so much valuable information that we would recommend running in temperature-ramping mode despite the obvious drawback of poorer time resolution.

It is possible that future improvement in the design of the VIA could limit the decomposition, and thereby make the quantification easier. Alternatively, the ramping could be done much faster if active cooling was introduced, in which case the thermogram information would remain, while still improving the time resolution. We added more discussions on the correction factor in Sec. 3.4.

The correction factor might be related to functional groups as the reviewer mentioned, but without information on real molecular-level measurements, we used molecular weight as the x-axis to show the general trend of the correction factor in Figure 8b.

**Comment #4:**

I would suggest the authors to briefly discuss the advantages and disadvantages in Sect. 3.4 compared with other techniques mentioned in the introduction part.

**Response:**

Section 3.4 "Current challenges and future improvements" was primarily aimed at discussing limitations of the VIA-NO$_3$-CIMS system and potential future hardware upgrades. A comparison to other techniques inevitably requires detailed knowledge about those techniques, for example, their sensitivity towards HOM. As the NO$_3$-CIMS was selected as the detector for this purpose in our study, we are not selective towards less oxygenated species, which a PTR or I-CIMS likely would be. Overall, we prefer to not make too explicit comparisons to how other instruments perform, but we did add a section highlighting the strengths and weaknesses of the VIA-NO$_3$-CIM more generally in Line 522.

*"In comparison to other online techniques used for aerosol phase characterization, the VIA-NO$_3$-CIMS has both benefits and drawbacks. The NO$_3$-CIMS was chosen due to its sensitivity and selectivity towards HOMs, which ultimately means that we can use it to measure OA composition with a low detection limit, but will not be able to detect all the evaporated species. This was particularly clear from the fact that we do not detect any of the decomposition products of the HOM, as they are going to be smaller and less oxygenated, whereby they do not readily cluster with the nitrate ions in our CIMS."*

**Comment #5:**

2b, in the legend, is "140 ºC" the set temperature?

**Response:**

All figures used the read temperatures (i.e. recorded by the Eyeon software). We added one sentence in line 150 to clarify this point.

*"A thermocouple attached to the surface of this vaporization tube was used to monitor the temperature, and the recorded temperature was used for thermogram analysis."*

**Comment #6:**

7b, is the normalize frequency of ΔT obtained from each molecular formula? Can the difference in chemical composition at different aerosol loading influence the distribution of the frequency?

**Response:**

Yes, the ΔT distributions were calculated based on the identified molecular formula. We used the same peak list during the high-resolution peak fitting process for all experiments. If there were peaks that showed very low signals (i.e. a "bad" shape of thermogram) in at least one experiment so that a reliable $T_{max}$ could not be obtained, the peaks were excluded from the statistics. We tried to compare the same peaks among several experiments with different SOA loadings. However, if the same molecular formula is in fact different compounds, this will affect the distribution, but this limitation is related to mass spectrometry in general.

**Comment #7:**

L433, Ren et al 2022 could be mentioned here.

**Response:**

The work by Ren et al. (2022) compared the effects of calibration method (syringe deposition vs. atomization) and matrix effects of inorganics on the volatility calibration of FIGAERO calibration. Thus, we cited this work followed reviewer's suggestion in line 433.

*"The calibration method (syringe deposition vs. atomization), solution/mass concentration, particle size, matrix effects of inorganics, and heating ramp rate are factors that have been reported to affect the determined volatility (i.e. Tmax) in the FIGAERO-iodide-CIMS system (Ylisirniö et al., 2021; Ren et al., 2022)."*

**Comment #8:**

L511, what does the "correlation coefficient" denote?

**Response:**

The correlation coefficient refers to the mass concentrations measured by the VIA-NO$_3$-CIMS system vs the SMPS/AMS. We acknowledge that this was poorly formulated and in order to better clarify this part, we modified the sentence in line 511.

*"On the one hand, the uncertainty of this correction factor could potentially be reduced significantly if the detection efficiency could be determined experimentally (i.e. mass concertation of standards measured by the VIA system vs. the SMPS)."*

**Reference:**

Ren, S., Yao, L., Wang, Y., Yang, G., Liu, Y., Li, Y., Lu, Y., Wang, L., and Wang, L.: Volatility parameterization of ambient organic aerosols at a rural site of the North China Plain, Atmos. Chem. Phys., 22, 9283-9297, https://doi.org/10.5194/acp-22-9283-2022, 2022.

---

## Author Comment (AC3)

**Response to Reviewer #RC2**

**General comment**

Organic aerosols are a major contributor to total aerosol mass concentrations and have implications for both human health and climate change. However, the formation of organic aerosols involves a variety of chemical and physical processes in the atmosphere, resulting in complex particle compositions. Therefore, the measurement and quantification of particle composition, especially at the molecular level, has been a long-standing measurement technology challenge that is critical for a better understanding of the sources and formation mechanisms of organic aerosols.

This paper presents an improved thermal desorption technique, the Vaporization Inlet for Aerosols (VIA), coupled to the $NO_3$-CIMS. The VIA inlet removes gas compounds with an activated carbon denuder, vaporizes particles in a heated tube, and transfers the thermally desorbed vapors to the $NO_3$-CIMS with a newly designed sheath flow interface. The authors demonstrate that the VIA inlet can efficiently remove background gas compounds while maintaining high transmission of particles larger than 50 nm, and that the sheath flow interface achieves low detection limits of desorbed vapors due to reduced wall loss. In addition, the authors show that the VIA inlet can also be operated in a temperature ramping mode, where the volatility of particulate compounds can be probed through thermogram analysis. The scientific topic of this paper is important, the measurement technique is novel, and the technique characterization is comprehensive. Overall, this is a relevant study that fits within the scope of the AMT. However, some technical details need further clarification and discussion to make it more useful to the community. Here are my main questions/comments:

We thank the reviewer for the positive and insightful comments, and we answer the specific comments point-by-point below. The reviewer's comments are in **blue**, and our answers are in **black**.

**Specific comments**

**Comment #1:**

While the VIA-$NO_3$-CIMS is an online technique when operating at a fixed T, it appears to have long duty cycles (hours) for T ramping. Are there limitations that prevent rapid ramping? If so, the authors should mention them in the main text, as volatility measurement is a key feature of this technique.

**Response:**

The main factor that limits the time resolution in the current setup is the time needed for cooling after a heating ramp. As noted in Section 3.4, at least 20-30 minutes are needed for its natural cooling down to room temperature. We also discuss the potential for active cooling in this section, which could increase the time resolution if implemented.

If the hardware does not introduce limitations to the time resolution, the next limitation comes when we start ramping rapidly enough that the shapes of the thermograms become noisy due to shorter data averaging at each temperature. At this point, the feasible maximum ramping rate becomes a function of the aerosol loading. Nevertheless, we expect that around one hour for a full cycle (up and down ramp) is feasible, and if data from both ramps can be used, this provides an effective time resolution of 30 minutes. We modified the discussions in line 508 to clearly point out the two factors that prevent rapid temperature ramping, i.e. the lack of a cooling system and enough data points to fit the thermograms with more accurately.

**Comment #2:**

When operating in T-ramping mode, whether particles are fully evaporated can be judged from the shape of the thermograms. However, when operating at a fixed T for high time resolution, it's less obvious to me how to tell if 0.1 s residence time is sufficient for complete evaporation, especially for aerosol loading in polluted environments. And this introduces quantification uncertainties into the online measurement. The authors should discuss this.

**Response:**

This is a very good point. For the application in field or laboratory settings where fast changes in aerosol loadings are expect, e.g. close to primary sources, it may be more useful measure at one fixed temperature (which can evaporate the most fraction of SOA particles). This does add uncertainty to the quantification but will provide better chemical information of rapidly changing aerosol components. A compromise could also be to periodically run one temperature scan to get the correction factors, e.g. twice a day. In all cases, having an AMS or SMPS system after the VIA during those scans at the same time would be very helpful to check if the particles evaporated completely and to constrain the fitting. Alternatively, to correct for fast changes in the aerosol loadings, and AMS/SMPS measuring ambient air can be used to normalize the VIA thermograms.

We added the following discussions in Section *"3.4 Current challenges and future improvements"* covering the limitations of application for field campaigns.

*"The application of the VIA-NO$_3$-CIMS system in the field will be one critical next step. In most cases, using the ramping mode is to be preferred, as the additional information from the thermograms aid both quantification and estimations of volatility. Fixing the thermal desorption temperature could be preferable under conditions where aerosol loadings are expected to change on short time scales, e.g. close to large primary emissions. In these cases, quantification is limited, but chemical information can be obtained from short-term plumes. As a compromise, most of the time measuring at one fixed temperature (which can evaporate the major fraction of OA), but running entire temperature scans routinely to get the correction factors, e.g. twice a day, could be an option. In addition, having an AMS or SMPS system after the VIA during scans at the same time would be very helpful to check if the particles evaporated completely. Though there would also be value in having an AMS/SMPS measuring ambient air all the time to provide information on how the total OA signal changes throughout a ramp. At some point it is clear that the sensitivity of the VIA system becomes the limiting factor in capturing very fast changes in the composition."*

**Comment #3:**

Thermogram analysis and the corresponding 1-D model are valid for a constant particle source. However, if the T ramp takes hours (or even 10s of mins), how would this technique account for variations in particle composition and size distribution for ambient measurements?

**Response:**

It is clear that in conditions where large changes in the aerosol types and loadings take place on the time scale of 10s of minutes, the VIA is not going to be optimal. We refer to our response to Comment #2 where we also addressed this issue.

**Comment #4:**

The authors attribute the decreasing HOM signals after reaching their maximums to the vapor wall loss in the vaporization tube. It's true that molecular diffusion, and thus wall loss, increases with temperature, but I'm not entirely convinced that this can cause > 90% loss as shown in Fig S14. Could this decrease also be thermal decomposition? The lack of double modes in the thermograms may simply be that the

decomposition products are less oxygenated, which escapes detection by NO3-CIMS. The authors would need to justify their conclusion.

**Response:**

The reviewer is correct that thermal decomposition is likely to play an important role, and the loss to the VIA walls was indeed assumed to be leading to decomposition and not solely being condensational loss. In fact, condensation is less likely as the VIA walls are hotter than the air in the VIA during the measurements, and if the molecules evaporate in the middle of the VIA, they should not stick to the warmer walls. The observed thermograms of sulfuric acid supported this hypothesis. In addition, we observed near-unit transmission of 13 tested VOCs within the thermal desorption tubing (Fig. S4 & S5).

Instead, as the reviewer speculated, there are chemical processes (possibly thermal decomposition) responsible for the vapor loss of these labile HOM molecules after they hit the hot wall. The main effect, leading to the steep decline in the thermograms is not mainly from diffusion increasing with temperature (although this also takes place), but the species evaporating earlier in the tube, thus having more time to diffuse to the walls.

We mentioned in the Abstract and the Conclusion that *"the loss potentially leads to fragmentation products that are not observable by the NO$_3$-CIMS."* To make this issue more clear, we also made modifications to indicate that the loss is likely chemical loss instead of the physical condensational wall loss in line 398, *"...higher temperatures might cause earlier evaporation within the vaporization tube, thus leading to larger losses (i.e. HOM vapors collide with the walls and decompose)."* and in line 465, *"We suspect this loss is mainly owing to that HOM molecules were lost upon impacting the walls of the vaporization tube, then decomposing to fragmentation products that are not detectable by the NO$_3$-CIMS"*.

Future work will aim to combine the VIA inlet with other detectors to also investigate the fates of less oxygenated compounds (and hopefully the thermal decomposition products as well) in SOA particles.

**Comment #5:**

If the vapor wall loss in the vaporization tube is indeed significant, this can introduce contaminations due to the wall memory effect when the VIA is cooled and heated again (e.g., Fig S10b). What level of quantification uncertainty would be introduced in the continuous operation of the T-ramping mode?

**Response:**

The comparable results from continuous steps down and ramping up runs in Figure S10 qualitatively indicates that the memory effects using the VIA-NO$_3$-CIMS is negligible. In order to quantify the memory effects of potential contaminations, we did one experiment in a 2 m$^3$ Teflon chamber. We included 3 background measurements: before, during, and after 12 continuous ramping scans (90 ppb of α-pinene and 62 ppb O$_3$).

In general, the relative contributions of contamination to the measured total signals are less than 2%, even smaller than the precision of the 12 scans (i.e., standard deviation, 4.6%). Thus, we attached both the 12 SOA measurement and 3 background scans of some HOM peaks in the supplementary (as Figure S14) to show the negligible background levels compared to the measurements. The following description was added in line 437 *"Reproducible thermograms were obtained using a steady SOA input and showed negligible background levels during continuous ramping mode (Fig. S14)."*

Overall, the contaminations and corresponding memory effects should not be a concern while using this setup (at least with SOA mass concentrations less than 40 µg m$^{-3}$ based on our tests).

[Figure]

*"Figure S14. Thermograms during SOA (green lines) and background (gray lines) measurements. The left panels are in linear scale, while the right panels are in log scale. The experiment was conducted in a 2 m³ Teflon chamber with 90 ppb of α-pinene and 62 ppb O₃. The three background measurements were conducted before, during, and after the 12 continuous ramping scans."*

**Comment #6:**

Fig 5b & c, why does the HOM trace in stepping mode seem less stable and smooth than in ramping mode?

**Response:**

At lower temperatures (30-150 $^{\circ}$C), both modes are relatively stable. However, when the temperature increases further up, the steps-mode measurements are not that flat anymore. In fact, the ramping-mode measurements start to show larger variations compared to lower temperatures as well, but less significant than the steps mode.

We suspect that this fluctuation might be related to the temperature-induced changes in the flows in the CIMS inlet. As shown in Fig. R1, the sum of reagent ions showed an opposite trend to the variations of the VIA temperature. As a result, normalizing the measured HOM signals to the total reagent ions could largely compensate for this fluctuation. This is partly why we did not include a part to explicitly discuss this fluctuation of reagent ions in the first place and we try to make the discussion relatively easy to follow and less redundant. Since normalization is usually the first step to do during the data analysis, we should have done the normalization for Fig. 5b & c as we did in other figures (e.g. Fig. 5a). Thus, we modified this figure in the manuscript accordingly and attached the revised version below. Now, the measurements of both modes are more comparable than before.

On the other hand, the temperature-dependent variation of the reagent ions (and total ion counts) during ramping mode are smoother than these step changes (blue curve in Figure R1). We found out that the mass loading of particles might also relate to this variation, but we do not have a clear explanation so far. Thus, we added some discussions on this issue in Section 3.4 "Current challenges and future improvements" to inform this variation of reagent ions during temperature changes to the reviewer and readers.

[Figure]

**Figure R1.** Time series of (a) reagent ions (i.e. sum of nitrate ion monomer, dimer, and trimer signals) and the VIA temperature, (b) raw (gray lines) and normalized (green lines) total HOM signals.

[Figure]

*"Figure 5. Comparison of measurements obtained between the ramping and the steps mode for (a) thermogram of some chosen HOM monomers and dimers, and (b, c) the relative contributions of different C number families to the total HOM signals. In (a), smoothed signals are shown for the ramping mode ("Loess" algorithm with a bandwidth of 0.25 and the second order local polynomial was used), while the mean (diamond markers) and standard deviation (bottom and top whiskers) are shown for the steps mode. The thermograms are normalized to the reagent ions first and then to their maximums. The raw signals of the same dataset are given in Fig. S10."*

**Reference:**

Lopez-Hilfiker, F. D., Mohr, C., Ehn, M., Rubach, F., Kleist, E., Wildt, J., Mentel, T. F., Lutz, A., Hallquist, M., Worsnop, D., and Thornton, J. A.: A novel method for online analysis of gas and particle composition: description and evaluation of a Filter Inlet for Gases and AEROsols (FIGAERO), Atmos. Meas. Tech., 7, 983-1001, 10.5194/amt-7-983-2014, 2014.

---

## Author Comment (AC4)

**Response to Reviewer #CC1, Ezra Wood**

**Comment**

Under what range of humidity values were the experiments that quantified how well the denuder removes gas-phase compounds (sections 2.1.1 and 3.1.1) conducted? Friedrich et al. (AMT, 13, 5739–5761, 2020, https://doi.org/10.5194/amt-13-5739-2020) demonstrated degraded performance of a similar activated carbon denuder to various nitrogen oxides under humid conditions compared to dry conditions. Ideally there is no humidity dependence for the denuder at hand to the range of organic compounds studied, but it would be reassuring for this to be experimentally determined.

**Response (our first response):**

Many thanks for this comment. We did not consider this humidity effect before. Our gas denuder experiments were conducted under dry conditions, and we observed near unity removal efficiency for the tested VOC, similar to those reported for nitrogen oxides with dry flows (Friedrich et al., 2020). But as reported by Friedrich et al. (2020), the performance of the gas denuder may decrease as the humidity increases, and in their experiments, the removal efficiency decreased to ~65% for some NOy species. Furthermore, a "used" denuder was found to release NOx (converted from stored NOz at the surface) in humid air (Friedrich et al. 2020). Testing for such effects also for organics would be very important for our study, and for other studies using a similar gas denuder as we used, e.g. in the chemical analysis of aerosol online (CHARON) inlet (Eichler et al., 2015) and the extractive electrospray ionization time-of-flight mass spectrometer (EESI-TOF) system (Lopez-Hilfiker et al., 2019). In addition, there are studies that reported similar degraded adsorption capability of activated carbon denuders to VOC in humid conditions, and this effect depends on the coating and design of the denuders (Li et al., 2020; Li et al., 2021).

In practice, dryers are typically used for ambient aerosol measurements, and thus potential humidity effects can be minimized even if the denuders show such behavior. But it is indeed very interesting and important to investigate the humidity-dependent removal efficiency of organic compounds for the denuder used in our study. We expect to be able to conduct such tests during the next months when we have access to the necessary instrumentation and will thus be able to give a more detailed answer still during this review process.

References:

Eichler, P., Müller, M., D'Anna, B., and Wisthaler, A.: A novel inlet system for online chemical analysis of semi-volatile submicron particulate matter, Atmos. Meas. Tech., 8, 1353-1360, 10.5194/amt-8-1353-2015, 2015.

Friedrich, N., Tadic, I., Schuladen, J., Brooks, J., Darbyshire, E., Drewnick, F., Fischer, H., Lelieveld, J., and Crowley, J. N.: Measurement of NOx and NOy with a thermal dissociation cavity ring-down spectrometer (TD-CRDS): instrument characterisation and first deployment, Atmos. Meas. Tech., 13, 5739-5761, 10.5194/amt-13-5739-2020, 2020.

Li, X., Zhang, L., Yang, Z., He, Z., Wang, P., Yan, Y., and Ran, J.: Hydrophobic modified activated carbon using PDMS for the adsorption of VOCs in humid condition, Separation and Purification Technology, 239, 116517, https://doi.org/10.1016/j.seppur.2020.116517, 2020.

Li, Z., Jin, Y., Chen, T., Tang, F., Cai, J., and Ma, J.: Trimethylchlorosilane modified activated carbon for the adsorption of VOCs at high humidity, Separation and Purification Technology, 272, 118659, https://doi.org/10.1016/j.seppur.2021.118659, 2021.

Lopez-Hilfiker, F. D., Pospisilova, V., Huang, W., Kalberer, M., Mohr, C., Stefenelli, G., Thornton, J. A., Baltensperger, U., Prevot, A. S. H., and Slowik, J. G.: An extractive electrospray ionization time-of-flight mass spectrometer (EESI-TOF) for online measurement of atmospheric aerosol particles, Atmos. Meas. Tech., 12, 4867-4886, 10.5194/amt-12-4867-2019, 2019.

**Response (our current response):**

During the past months, we had a chance to conduct some further experiments to assess the performance of the gas denuders under humid conditions. A Vocus proton transfer reaction (PTR), equipped with a long time-of-flight mass spectrometer, and the same VOC cylinder as described in section 2.1.1 were used during the experiments. We tested the performance of two different gas denuders under dry (<1%) and humid (73$\pm$2 %) conditions, as the best and worst scenarios, respectively, with a working flow rate of 1 L min$^{-1}$. Note that 13 different VOC species with a concentration of ~10 ppb for each specie (thus ~130 ppb in total, which is much higher than ambient conditions) was used during the experiment. But, we only plotted alpha-pinene as an example, because they all showed comparable variations during the experiment.

The results of this experiment are shown in Figure S5 and summarized in Table S4, which have been added to the revised manuscript. In general, better removal efficiency was obtained under dry compared to humid conditions for both the "new" and "old" gas denuders (i.e. lower concentrations of alpha-pinene during the "VOC + denuder" stages in light red area vs. in light blue area). In addition, during the continuous exposure experiment (Figure S5b), we observed faster performance degradation of the gas denuder under humid compared to dry conditions, indicating extra caution is needed during long-term usage of these gas denuders.

Overall, as the reviewer pointed out that the humidity effect is very important, we added more discussions in Section 3.1.1 line 274, to inform the readers about the results of our recent experiment.

**"Table S4.** *Description and performance of the two tested gas denuders. The time series are shown in Figure S5.*

| Gas denuder | #1 | #2 |
|---|---|---|
| Description | | |
| Usage count | > 20 | 1 |
| Regeneration count | > 20 | 1 |
| Status before test | regenerated | regenerated |
| Usage time | ~5 month | ~1 week |
| Description | "old" | "new" |
| Performance (removal efficiency in terms of ~10 ppb alpha-pinene) | | |
| Dry | 87% | 94 % |
| Humid | 76 % | 84 % |
| continuous (~18-20 h) exposure to VOC under | | |
| Dry | / | 92% to 87 % |
| Humid | 92% to 81% | / |

*"*

"

[Figure]

***Figure S5.*** *Time series of alpha-pinene during (a) gas denuders test under dry (light blue area) and humid conditions (light red area) and (b) continuous exposure to VOC under dry and humid conditions. Because the rest VOC standards showed comparable variation during this experiment, alpha-pinene is plotted here as an example to show the humidity effects on the performance of gas denuders. In panel (a), the text explains the condition of each stage, i.e. different combinations of VOC flow and gas denuder, and strikethrough means without. The performance of these two gas denuders is summarized in Table S4.*"